# EVALUATING EXPERT SPECIALIZATION IN MIXTURE-OF-EXPERTS ANTIBODY LANGUAGE MODELS

**Sarah M. Burbach, Simone Spandau,** [*] **Jonathan Hurtado & Bryan Briney**
Department of Immunology and Microbiology
The Scripps Research Institute, USA

Correspondence: `briney@scripps.edu`

## ABSTRACT

Antibody language models (AbLMs) show an impressive aptitude for learning antibody features, but tend to struggle learning the highly diverse, non-templated regions of antibodies. Existing AbLMs use dense architectures, where all model parameters attend to each amino acid token. We hypothesized that the modular nature of antibodies could benefit from a sparse mixture-of-experts (MoE) architecture, allowing specific parameters (referred to as 'experts') to specialize in distinct antibody features. While MoE architectures are widely adopted and optimized in natural language processing domains, they are less common in biological modeling. To this end, we assess existing MoE routing strategies and find that token-choice routing strategies outperform expert-choice routing, presumably due to their specialization in CDRH3 residues. We further optimized the token-choice router for AbLMs, by minimizing the routing of padding tokens to enable pre-training with varying sequence lengths. Finally, we show that a large-scale baseline antibody language model with a Top-2 MoE architecture (BALM-MoE), trained on a mixture of unpaired and paired antibody sequences, outperforms its dense counterpart with the same number of active parameters.

## 1 INTRODUCTION

The development of the transformer architecture (Vaswani et al., 2017) in 2017 sparked a flurry of work training language models (LMs) for natural language processing (NLP). Much like the way that the words in a sentence determine its meaning, the order and context of amino acids in a protein sequence determine its function and structure. Protein LMs (pLMs) have been successful and widely utilized, particularly for their structure prediction abilities (Lin et al., 2023; Hayes et al., 2025). However, pLMs typically struggle with antibodies (Olsen et al., 2022b), due to limited exposure to antibodies in training data and the unique characteristics of antibody sequences.

Antibodies are highly diverse, enabling them to bind to a large breadth of antigens. This diversity is first established via somatic recombination of modular germline gene segments, which is estimated to produce as many as $10^{18}$ unique antibody sequences (Briney et al., 2019). Heavy chains recombine variable (V), diversity (D), and joining (J) segments, while light chains recombine only V and J segments. These sequences are matured to target specific antigens after exposure, by a process called affinity maturation (Tonegawa, 1983). Mutations are driven in the complementarity-determining regions (CDRs), which are typically the site of antigen contact. This results in mature antibodies that are primarily germline gene segments with a few functionally critical mutations.

Specialized antibody LMs (AbLMs) have been shown to better learn these nuances than pLMs (Leem et al., 2022; Olsen et al., 2022b). Despite this, the modular nature of antibody assembly poses a serious problem for AbLMs attempting to learn the rules governing antibody diversity. Since antibody sequences overwhelmingly encode germline residues, models struggle to learn non-templated regions and mutated positions that deviate from the germline (Olsen et al., 2024). This is most prominent in the CDR3 of the heavy chains (CDRH3), the most diverse region of the antibody sequence. This is a significant issue, as models with severe germline bias will be substantially less

---

[*] Current address: Department of Computer Science and Engineering, University of California San Diego

useful for tasks like antibody engineering and simulating affinity maturation. Recent work has attempted to mitigate germline bias using data-driven approaches such as focal loss (Lin et al., 2017; Olsen et al., 2024) or preferential masking (Ng & Briney, 2025), to encourage more efficient learning of non-templated regions.

We hypothesized that an architecture-driven approach could offer an alternative means of addressing germline bias. Specifically, existing AbLMs typically use "dense" architectures, meaning every model parameter is applied to every input token. In contrast, mixture-of-experts (MoE) models are "sparse" architectures that dynamically select which parameters should be activated for a given input. This results in a lower number of active parameters relative to the total parameters in the model, allowing for very efficient training and inference of large LMs (Shazeer et al., 2017). For transformers, this is usually accomplished by replacing the feedforward layers of some or all transformer blocks with MoE layers (Fedus et al., 2021; Rajbhandari et al., 2022). In addition to their compute efficiency, MoE models generally outperform their dense counterparts with a similar number of active parameters (Fedus et al., 2021; Jiang et al., 2024). This performance boost has been attributed to expert specialization, in which individual experts preferentially attend to a subset of input tokens with similar properties (Zoph et al., 2022; Chen et al., 2022). Expert specialization is heavily influenced by the mechanism by which tokens are routed to experts. Existing routing strategies include expert-choice (EC) routing (Fedus et al., 2021) (where each expert chooses its tokens of interest) and token-choice routing (where each token chooses its expert(s) of interest). There are variations of token-choice routing, including Top-K routing (Lepikhin et al., 2020; Fedus et al., 2021) where each token chooses 'k' experts and Top-P routing (Huang et al., 2024) where each token is routed to experts until a threshold is reached.

MoE architectures are widely adopted in NLP and are used in the most groundbreaking language models, such as the DeepSeek (Dai et al., 2024; DeepSeek-AI et al., 2024), Qwen (Yang et al., 2024; 2025), and Kimi (Kimi Team et al., 2026) models. There has been some success using MoE architectures for pLMs (Sun et al., 2024), suggesting potential for their application to antibodies. We theorized that the expert specialization of MoE architectures may be particularly useful for handling the imbalance between germline and non-germline residues. In particular, we were interested in how different routing strategies would influence expert specialization in sparse AbLMs. In this study, we first explore the baseline performance of three common routing strategies (EC, Top-K, and Top-P) and then optimize them for AbLMs. We show that MoE architectures, specifically a Top-K router with minimal modifications, outperform their dense counterparts, showing promise for further mitigating the AbLM germline bias problem as dataset sizes continue to scale.

## 2 RESULTS

### 2.1 TRAINING PILOT MoE AbLMs

To assess existing sparse architectures, we tested a variety of pilot-scale model configurations. Pilot MoE models were encoder-only models containing 45M active parameters, with varying numbers of total parameters. MoE router implementations were modeled closely on previous implementations (Fedus et al., 2021; Lepikhin et al., 2020; Zhou et al., 2022; Huang et al., 2024). For the token-choice routers, one major change was implemented to add a sorting of tokens by probability before applying the expert capacity, ensuring that the highest probability tokens are seen by each expert. See the methods for more details about the model architecture and training setup.

Previous studies on AbLMs have shown that paired sequences (containing both the heavy and light chain) are superior compared to unpaired sequences (containing only a single chain) for model training (Burbach & Briney, 2024; Kenlay et al., 2024a; Olsen et al., 2024). However, paired sequences are limited and therefore limit model scale (Neyestanak et al., 2025). In addition, mixing unpaired and paired sequences requires more complex training strategies to prevent issues such as catastrophic forgetting (Kenlay et al., 2024a; Burbach & Briney, 2025). Therefore, all pilot models are trained exclusively on unpaired sequences from the Observed Antibody Space (OAS) (Olsen et al., 2022a). Models were trained for 250k steps, equating to 0.4 epochs of the unpaired dataset.

## 2.2 TOKEN-CHOICE ROUTERS OUTPERFORM EXPERT-CHOICE ROUTERS ACROSS MODEL CONFIGURATIONS

First, we explore how three routing strategies (EC, Top-P, and Top-K) perform as AbLMs (Fig 1A). For Top-K models, we set k=2 for initial tests. Beyond the routing function itself, there are many possible hyperparameters to consider, two of which are especially interesting for AbLMs: expert capacity (Fedus et al., 2021) and shared experts (Rajbhandari et al., 2022). Expert capacity limits the number of tokens routed to each expert and is defined using a multiplier (see Appendix A.1, Equation 1). A shared expert receives all tokens in the batch, regardless of routing. To explore the impact of these hyperparameters, we trained 6 variations of each router across three expert capacity multipliers (0.25, 0.5, and 1.0), with and without a shared expert, while jointly scaling the expert feed-forward network (FFN) dimension to maintain 45M active parameters (Fig 1B). For comparison, we also trained a dense model with 45M total parameters.

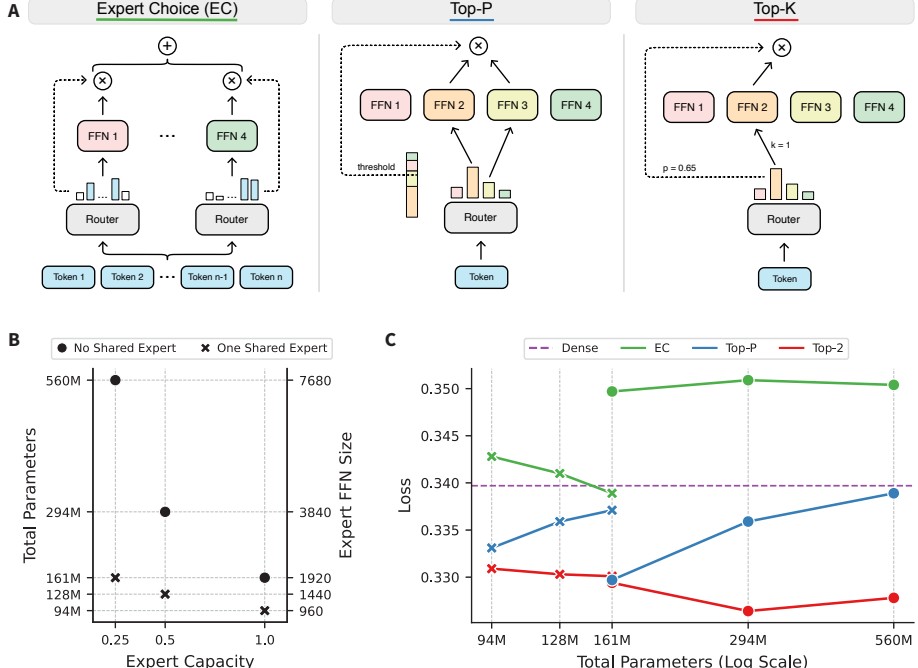

Figure 1: Comparing existing MoE routing strategies with varying expert capacities and shared experts. (A) Routing schematics for three existing MoE routing strategies: EC, Top-P, and Top-K. (B) Hyperparameter configurations for six MoE models, where the number of shared experts, expert capacity, and expert FFN dimension were jointly adjusted to maintain 45M active parameters. Each routing strategy was trained with these configurations, for a total of 18 models. (C) Cross-entropy loss on 100k heavy chains randomly sampled from the test dataset, plotted against the total model parameters. Loss for the control dense model is represented with a dashed purple line.

We evaluated these models with a masked language modeling (MLM) objective on 100k heavy chains from the test set (Fig 1C). Results for the same task on 100k light chains, which show similar trends, can be found in Appendix A.2, Fig 5. Regardless of expert capacity and shared experts, the Top-2 models consistently outperform the other routing strategies. Differences in performance between the Top-2 and Top-P models are smaller, but the EC models greatly underperform the token-choice and even the dense control model. The only exception is the EC model with an expert capacity of 0.25 and a shared expert, which is the only EC model to outperform the dense model.

However, this configuration preference appears limited to the EC models. For instance, the token-choice models appear to benefit less from a shared expert compared to the EC models. Top-P models also prefer a less restrictive expert capacity, with the best performing model having an expert capacity of 1.0. This is expected because Top-P routing was designed without an expert capacity,

allowing tokens to route to as many experts as needed until a threshold is reached. With that said, changes in the expert capacity and shared experts are linked with changes in the total parameters; this means it is difficult to determine the cause of the observed trends. In addition, models were trained for the same number of steps, which is a disadvantage for larger models that converge slower. However, this does not affect comparisons between routers of the same size, since we observe that trends emerge early in training and remain consistent across checkpoints (Appendix A.2, Fig 6).

## 2.3 SPECIALIZATION IN TOKEN-CHOICE ROUTING IMPROVES PREDICTIONS IN THE DIVERSE CDRH3 REGIONS

To further examine the trends between routers observed in Fig 1, we assessed model performance on the diverse CDRH3 regions. To minimize complications of evaluating models of varying sizes, we assessed one group of models: those without a shared expert and the least restricted expert capacity of 1.0, containing 161M total parameters. We performed per-position masking of 5k heavy chains randomly sampled from the test set and observed that the Top-P and Top-2 models had significantly lower loss in the CDRH3 regions than the EC model (Fig 2A).

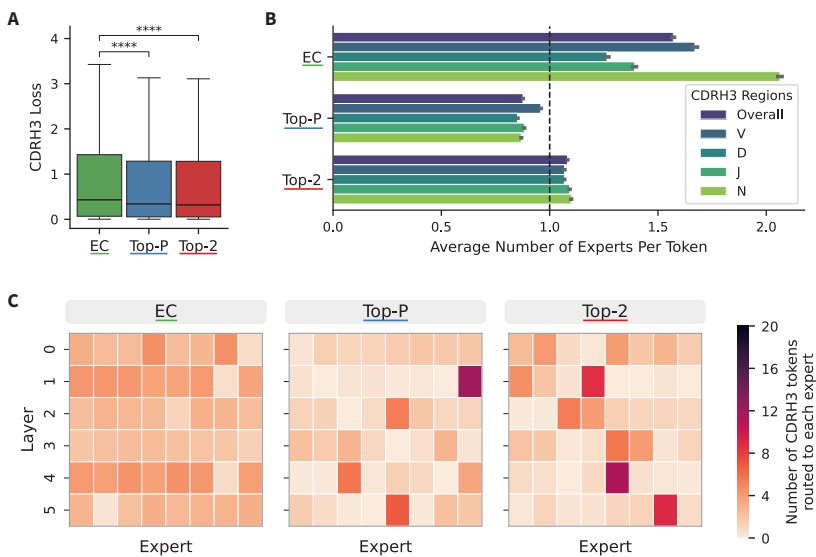

Figure 2: CDRH3 cross-entropy loss and routing behavior by router type. (A) Median cross-entropy loss on the CDR3 regions in 5k heavy chains from the test dataset. Statistical significance is calculated using a paired two-sided t-test with Bonferroni correction for three comparisons. (B) Average number of experts per token by region and overall within the CDRH3. (C) Heatmaps depicting the number of CDRH3 tokens routed to each expert across sparse layers. The maximum number of tokens routed to a given expert is 20, as indicated by the colorbar.

To explore the cause of this performance difference, we analyzed the routing of CDRH3 tokens to experts. First, we examined the number of experts CDRH3 tokens are routing to (overall and broken down by region within the CDRH3) (Fig 2B). The EC model routes the most CDRH3 tokens, assigning each token to ~1.5 experts on average. The high level of CDRH3 token routing is concentrated in the N-addition region, where each token is routed to ~2 experts on average. This increased routing aligns with the increased difficulty of the N-addition region, since N-additions are mostly random and therefore very difficult to predict. However, this trend is not observed in the token-choice routers. The Top-P model routes CDRH3 tokens to slightly less than 1 expert, while Top-2 routes each token to slightly more than 1 expert. This suggests that the improvement in CDRH3 performance is not linked to an increase in expert parameters devoted to this region.

Next, we evaluated expert specialization by plotting how many CDRH3 tokens route to each expert within each layer (Fig 2C). In the EC models, CDRH3 tokens are routed across the experts without any clear specialization. However, the token choice models (Top-P and Top-K) show higher levels

of specialization, with one or two experts receiving a majority of CDRH3 tokens in each layer. This suggests that the increased performance observed in the token-choice models is related to the specialization of experts. A similar, though less pronounced, pattern is observed in the CDR3 of the light chains (Appendix A.2, Fig 7).

## 2.4 OPTIMIZING ROUTING OF TOP-K MODELS IMPROVES THEIR PERFORMANCE ON MASKED LANGUAGE MODELING

While the token-choice routers show promising specialization in the CDRH3 region, further analysis of routing patterns reveals that these models route a significant number of pad tokens (Fig 3A). The percentage of total tokens routed that are pad tokens is highest in the Top-P and lowest in the EC models, and is not reduced by decreasing the expert capacity. Looking more closely at the pad tokens routed by the Top-2 model, the pad tokens routed are spread across the sequence length (Fig 3B). This suggests that the model isn't learning any particular pattern and is simply routing the tokens to fill the expert capacity. This is a concern because padding is necessary for training AbLMs, especially for training with a mixture of unpaired and paired sequences, where unpaired sequences are over 50% padding.

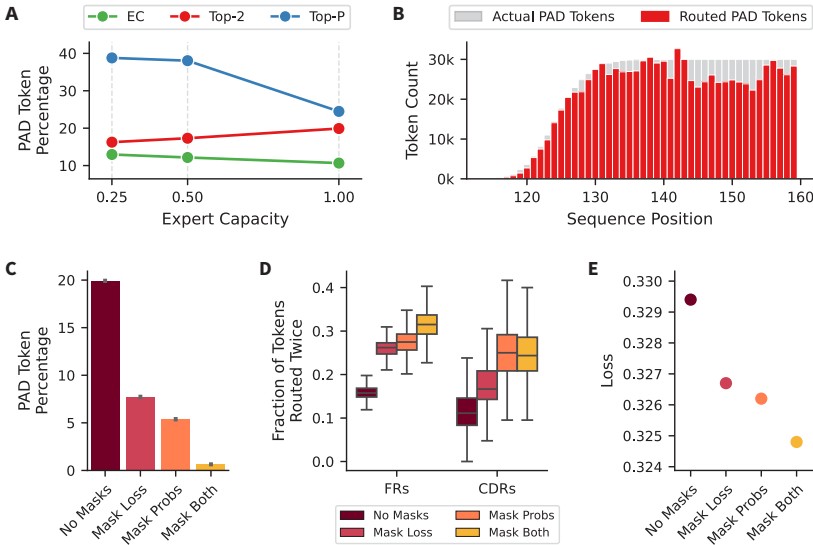

Figure 3: Reducing pad token routing in Top-2 models. (A) Percentage of routed tokens that were pad tokens, across router types and expert capacities, without a shared expert. (B) Routed pad tokens by sequence position in the Top-2 model with an expert capacity of 1.0. (C) Percentage of total tokens routed that were pad tokens for Top-2 models trained with and without the auxiliary loss and router probabilities masked. (D) Fraction of tokens routed twice in the FR and CDR regions, by model. (E) Loss on 100k heavy chains from the test set, for each model.

Therefore, we implemented several methods to discourage routing of pad tokens during training. First, we excluded pad tokens from the auxiliary loss, so that pad tokens don't contribute to the expert load balancing, preventing the occurrence of a specific pad token expert. Second, we minimize the routing probabilities of pad tokens in the router before sorting, resulting in many of the pad tokens being dropped when the expert capacity is applied. To test these two methods, we trained 3 models: one with the aux loss masked ('Mask Loss'), one with the probabilities masked ('Mask Probs'), and one with both masked ('Mask Both'). Comparing these models to the standard Top-2 model ('No Masks'), we observe that masking one of these reduces the percentage of pad tokens, but the combination of both reduces it the most (Fig 3C). In conjunction, we observe that the reduction of pad tokens frees up the parameters of the experts to be devoted to more meaningful information. Specifically, we observe models increasingly routing the FR and CDR tokens more than once (Fig 3D). These changes appear to correlate with a reduction in the loss of heavy chain sequences as

well, with the 'Mask Both' model showing the overall cross-entropy loss on a dataset of 100k heavy chains sampled from the test dataset (Fig 3E). This suggests that adopting a strategy to reduce the routing of pad tokens is very useful for training MoE architecture AbLMs.

## 2.5 A LARGE-SCALE TOP-2 MODEL, BALM-MoE, OUTPERFORMS ITS DENSE COUNTERPARTS

While previous models in this paper were trained with only unpaired sequences, mixed-data models trained on both unpaired and paired sequences are far more common in recent models (Olsen et al., 2024; Kenlay et al., 2024b; Barton et al., 2024). Based on our findings thus far, we trained a 200M active parameter (710M total parameter) Top-2 MoE model on a mixture of unpaired and paired data, which we will call BALM-MoE. We utilized both methods for reducing pad token routing and trained with a constant mix (62.5% unpaired) of data types throughout training (Burbach & Briney, 2025) to stabilize the number of pad tokens per batch. We also trained two dense models, one with 200M parameters (Dense-200M) and one with 710M parameters (Dense-710M), both using identical training strategies for comparison.

Prior to evaluating the performance of these models, we assessed how the routing is affected by training on a mixture of data types. We evaluated BALM-MoE on three datasets of 5k paired, heavy chain, and light chain sequences from the test dataset, with all sequences padded to 256. First, we observe that almost no pad tokens are routed in the paired sequences, while 5-15% of the tokens routed in the unpaired heavy and light chains are pad tokens (Fig 4A). However, a closer look reveals that non-special tokens in the unpaired sequences are still routed twice in 80–90% of cases (Fig 4B). This suggests that the minimal routing of pad tokens is unlikely to interfere with the model's ability to attend to meaningful tokens. Second, we compared which experts CDRH3 tokens were routed to in paired versus unpaired heavy chain sequences (Fig 4C). We observe a strong positive correlation (r=0.916), indicating that CDRH3 tokens are being routed to similar experts in each layer regardless of the data type. This suggests that the model understands the similarities between the residues, regardless of whether the light chain is present.

To assess model performance, we computed inference on 100k paired, heavy unpaired, and light unpaired sequences from the test set (Fig 4D). We observe that BALM-MoE has a lower cross-entropy loss than both dense models across all three data types. The lower loss compared to Dense-710M suggests that the specialization achieved with Top-2 routing enables a real improvement in model performance on MLM tasks. As expected, BALM-MoE is also more computationally efficient than Dense-710M, running 2.4x more samples per second when performing inference on paired sequences using a single L40S graphics processing unit (GPU) (Fig 4E).

We additionally evaluated these models on downstream classification tasks, intended to assess the models understanding of antibody specificity. We performed two tasks: a binary classification with Healthy-Donor (HD) sequences and Coronavirus (CoV) specific sequences, and a three-way classification with HD, CoV specific, and Influenza (Flu) specific sequences (Fig 4F). In both tasks, we observe that BALM-MoE outperforms the Dense-200M model, but underperforms compared to the Dense-710M model across metrics. This result is expected because the Dense-710M model has a larger embedding dimension (1280 vs 640), creating more trainable parameters in the classification head. This suggests that, while MoE models perform well on MLM tasks, some of these gains may be lost in tasks containing trainable classification heads. This is an inherent trade-off when using an MoE model, because the smaller embedding dimension makes the model more compute efficient, but may slightly decrease its utility in downstream tasks.

## 3 DISCUSSION

Antibodies are modular and, even in mutated sequences derived from memory B cells, overwhelmingly encode germline residues, meaning some parts of the antibody are less informative while others are very diverse and difficult for models to learn. Previous studies have attempted to address these issues with alterations to the training process, such as focal loss and preferential masking, to focus the model on complex, information-dense regions of the antibody. We hypothesized that an architectural approach may aid in tackling this challenge. Existing models are dense architectures, meaning all tokens are processed by all model parameters. Here, we systematically tested the application of

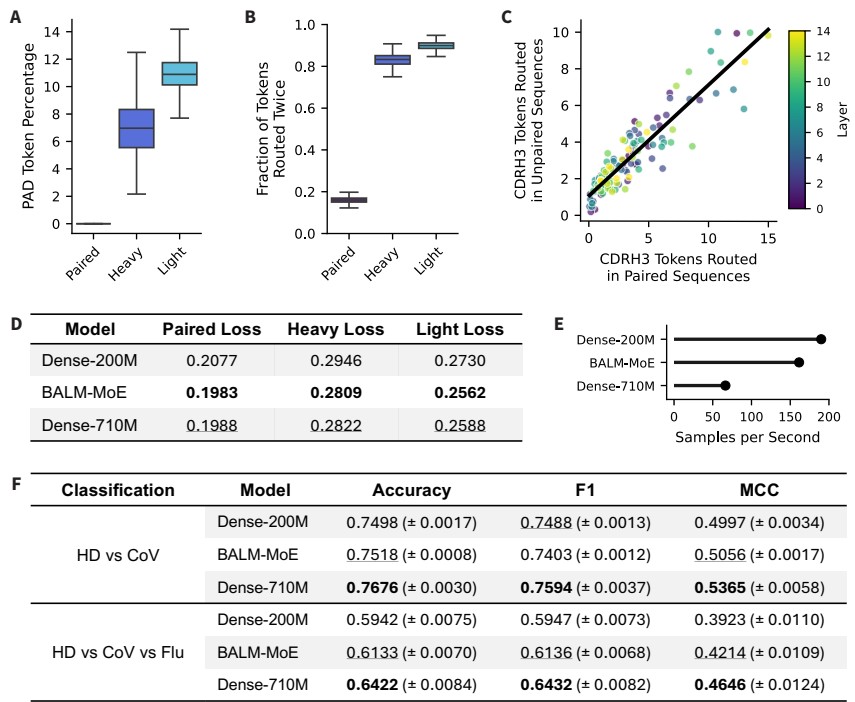

Figure 4: Evaluating BALM-MoE compared to size-matched dense models. (A) Percentage of routed tokens that were pad tokens on the three datasets randomly sampled from the test datasets: 5k paired sequences, 5k heavy chain sequences, and 5k light chain sequences. (B) Fraction of tokens routed twice by BALM-MoE on the three test datasets. (C) Scatterplot of the number of CDRH3 tokens BALM-MoE routed in paired versus unpaired heavy chain sequences. Each point represents a single expert-layer pair. The regression line has a Pearson's correlation of r=0.916. (D) Cross-entropy loss on three datasets randomly sampled from the test datasets: ~50k paired sequences, 100k heavy chain sequences, and 100k light chain sequences. (E) Samples per second of each model on paired inference performed on a single L40S GPU. (F) Results for two specificity classification tasks, Healthy Donor vs CoV and Healthy Donor vs CoV vs Flu.

sparse MoE architectures, which are computationally efficient models capable of specialization, to AbLMs.

We first assessed three existing routing techniques (Top-2 and Top-P, and EC), with a variety of expert capacities, and varying presence of a shared expert. We observed that token-choice routers, especially Top-2 routers, outperform EC routers. This performance difference is significant in the highly diverse CDRH3 region, likely due to specialization observed in the token choice routing. Interestingly, despite allocating the most compute to the more difficult regions of the CDRH3, such as the N-additions, the EC model performed the worst on this region overall. This suggests that specialization is more important for improving predictions than simply increasing the number of parameters.

Despite this initial assessment of MoE architectures, we also observed that routers route a significant amount of pad tokens. This is not ideal, because it allocates compute to essentially meaningless tokens. We implemented a method to reduce routing of pad tokens by excluding the pad tokens from the auxiliary loss and by masking the routing probabilities to prioritize non-pad tokens when the expert capacity is applied. We observed that the combination of these techniques reduces the routing of pad tokens to almost zero percent and lowers the overall loss of the model. This emphasizes the importance of adapting model architectures to the domain at hand.

This fix also allows for efficient training with a mixture of unpaired and paired sequences, where the unpaired sequences will be at least 50% pad tokens. To test this, we trained BALM-MoE, a

Top-2 MoE model with 200M active parameters, pre-trained on a mixture of unpaired and paired sequences. We also trained two dense models, with 200M and 710M parameters, to match the number of active and total parameters in BALM-MoE. Analysis of BALM-MoE routing patterns showed low routing of pad tokens and evidence that the model learned to route CDRH3 tokens to the same experts, regardless of the format of data supplied (that is, unpaired or paired). BALM-MoE also consistently outperformed dense models on the training objective of masked language modeling, on both paired and unpaired sequences. On downstream specificity classification tasks, BALM-MoE achieved a higher classification accuracy than Dense-200M, but was less accurate than Dense-710M, likely due at least in part to the smaller hidden dimension in BALM-MoE compared to Dense-710M.

These results show promise for the use of MoE architectures for AbLMs. However, the scale at which most AbLMs are currently trained (ie. ~350M to 650M parameters) is gated by the size of available training datasets; therefore, there are still trade-offs to consider between dense and sparse architectures. Since the limited data available limits model scale (Neyestanak et al., 2025), it is still computationally feasible to train the larger Dense-710M to get comparable MLM performance and stronger downstream task performance. In NLP, scaling laws established that performance improves with scale (Kaplan et al., 2020; Hoffmann et al., 2022) and MoE architectures were developed to adhere to this scaling law without significantly increasing the required compute (Cai et al., 2024). This suggests that the performance benefits of MoE architectures for AbLMs will increase as paired training dataset size scales and the computational burden of training AbLMs scale with it. In addition, a paired MoE AbLM may be very useful for structure prediction models, by reducing the memory requirements of the base model since structure modules are so large.

Further comprehensive evaluation of hyperparameters and training schedules could enable significant performance improvements for MoE AbLMs. For instance, evaluating the utility of different-sized experts may enable improvement on CDRH3 predictions. In addition, integration of MoE architectures with other domain-specific techniques to focus on the CDRH3 regions is likely to enhance performance.

In summary, our findings support the use of MoE architectures for AbLMs, particularly as datasets grow and minimizing computational cost becomes a bigger concern. For the use of MoE architectures, we observe that specialization, rather than a simple increase in the amount of parameters, is the driving factor of improved CDRH3 predictions in token-choice models. Additionally, we find that adapting model architecture to the domain, in this case by reducing the routing of pad tokens, improves model performance and enables training of mixed-data models, which is essential for training a state-of-the-art AbLM.

## 4 METHODS

### 4.1 DATASETS

Unpaired and paired sequences were downloaded from the OAS (Olsen et al., 2022a) in July 2025. The paired dataset was supplemented with 2.3 million internal sequences, primarily sequenced from memory B-cells in a high-throughput fashion using UDA-seq (Li et al., 2025). Sequences were annotated using abstar (Briney & Burton, 2018) and clustered using MMseqs2 at a 90% identity threshold. This resulted in a total of 159,443,126 unpaired sequences and 2,652,766 paired sequences. For both datasets, 96% of the sequences were randomly selected for model training, while 2% were excluded for evaluation, and another 2% for testing. For inference tasks, subsets of either 100k or 5k sequences were sampled from the test datasets.

For the specificity classification tasks, CoV-specific sequences were downloaded from CoV-AbDab (Raybould et al., 2021) on November 11th, 2024 and an internal donor L1236, Flu-specific sequences were obtained from Wang et al. (2023), and healthy donor sequences were obtained from the memory B-cell sequences of 5 donors in Neyestanak et al. (2025). Datasets were clustered at a 95% identity threshold and sub-sampled to contain an equal number of each class, resulting in a final dataset of 25,322 sequences for the HD-CoV classification and 3,672 sequences for the HD-CoV-Flu classification. Datasets were split for 5-fold cross-validation with stratification.

### 4.2 Model Architecture and Pre-training

Dense and MoE models were modeled after BERT-style encoder-only transformer architectures, with pre-norm, rotary positional embeddings (Su et al., 2021), and SwiGLU (Shazeer, 2020) activations. Models were written using PyTorch (Ansel et al., 2024) and trained using the Hugging Face (Wolf et al., 2019) and Accelerate libraries, with Deepspeed Stage 0 and BF16 mixed precision as in Switch Transformers (Fedus et al., 2021). Logging was performed with Weights & Biases.

MoE router implementations closely followed the source implementations. The expert choice router was based on Zhou et al. (2022). The Top-K router was based on the Switch Transformers (Fedus et al., 2021), GShard (Lepikhin et al., 2020), and DeepSeekMoE (Dai et al., 2024) implementations. The Top-P router was based on Huang et al. (2024). In the token choice routers (Top-K and Top-P), there was one main deviation from previous implementations. We added a sorting of tokens by probability before the expert capacity is applied, which ensures the highest probability tokens are routed and allowed us to reduce routing of pad tokens by zeroing out their probabilities.

Models use a ESM-2 vocabulary of 32 tokens, containing the 20 standard amino acids and other special tokens. We used the using the beginning-of-string token $<$cls$>$ as the separator token between chains. In pilot models, sequences were padded to 160 tokens (truncating only 125 sequences in the unpaired training dataset). In final mixed models, all sequences were padded to 256 tokens (truncating only 170 sequences in the paired training dataset).

More details about model configurations and training schedules can be found in Appendix A.1.

### 4.3 Model Evaluation

Inference was performed by masking a random 15% of residues and excluding any special tokens in the cross-entropy loss calculation. Per-position CDRH3 inference was performed by iteratively masking one residue at a time and calculating the median loss across the region. MoE routers were implemented with a flag to output expert token indices, to enable routing analysis.

For both specificity classification tasks, models were finetuned with a standard classification head with the base model parameters frozen. Evaluation metrics for the classification tasks are accuracy, F1, and Matthews correlation coefficient (MCC), calculated using scikit-learn (Pedregosa et al., 2011). Metrics were averaged across 5 cross-validation folds.

Plots were generated with seaborn (Waskom, 2021) and matplotlib (Hunter, 2007).

### 4.4 Code and Data Availability

The code is available on GitHub (`https://github.com/brineylab/MoE-paper`), the datasets are available on Zenodo (`https://doi.org/10.5281/zenodo.19411192`), and the model weights are available on Hugging Face (`https://huggingface.co/collections/brineylab/moe-paper`).

## 5 Funding

This work was funded by the National Institutes of Health (P01-AI177683, U19-AI135995, R01-AI171438, P30-AI036214, and UM1-AI144462) and the Pendleton Foundation.

## 6 Author Contributions

SB, SS and BB conceptualized the study. Data generation was performed by JH. Model code was written by SB, SS and BB. Model training and evaluation was performed by SB. The manuscript was prepared, revised, and reviewed by all authors.

## 7 Declaration of Interests

BB is an equity shareholder in Infinimmune and a member of their Scientific Advisory Board.

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

# A APPENDIX

## A.1 TRAINING DETAILS

Model parameters can be found in Table 1. All MoE models alternated dense and sparse layers, with each sparse layer containing 8 experts. For Top-K models, we set k = 2, and for Top-P models, we set the threshold to 0.7. This parameters could be further optimized for AbLMs in future work.

Table 1: Model parameters for dense and MoE models.

| Model | Active Params | Total Params | Layers | Heads | Hidden Dim | FFN Dim |
|---|---|---|---|---|---|---|
| Dense-45M | 45M | 45M | 12 | 20 | 480 | 1920 |
| Pilot MoE Models | 45M | 160M | 12 | 20 | 480 | 1920 |
| Dense-200M | 200M | 200M | 30 | 20 | 640 | 2560 |
| BALM-MoE | 200M | 710M | 30 | 20 | 640 | 2560 |
| Dense-710M | 710M | 710M | 27 | 20 | 1280 | 5120 |

Initial models were trained for 250k steps with a total batch size of 256 on 4 L40S GPUs. This equates to ~0.4 epochs of the unpaired dataset. These models were trained with 15,000 warm-up steps and a peak learning rate of 1e-4.

Large-scale mixed-data models were trained for 500k steps with a total batch size of 512 on 8 B200 GPUs. This equates to ~1 epoch of the unpaired dataset and ~39 epochs of the paired dataset. These models were trained with 30,000 warm-up steps and a peak learning rate of 1e-4.

The expert capacity was set using a multiplier, such that the number of tokens routed to each expert is dependent on the number of tokens in the batch and the number of experts, as shown in Appendix Equation 1.

$$\text{Expert Capacity} = \frac{\text{Multiplier} \times \text{Num Tokens per Batch}}{\text{Num Experts}} \tag{1}$$

All MoE models were trained with Z-loss (Zoph et al., 2022), to improve stability by encouraging small router logits. Top-K models were trained with auxiliary load balancing loss (Fedus et al., 2021; Jiang et al., 2024), to balance routing across experts. Top-P models were trained with a dynamic router loss (Huang et al., 2024), to discourage over-routing tokens. Router losses and their coefficients are indicated in Table 2.

Table 2: Loss coefficients during training. Checkmarks indicate that the loss was applied for a given router.

| Router Loss | Coefficient | Top-K | EC | Top-P |
|---|---|---|---|---|
| Z-loss | 0.001 | ✓ | ✓ | ✓ |
| Auxiliary | 0.01 | ✓ | | |
| Dynamic | 0.0001 | | | ✓ |

Downstream classification models were trained using BF16 mixed precision for 5 epochs, with a warm-up ratio of 0.1, and a peak learning rate of 5e-5. The binary classification (HD vs CoV) was trained with a total batch size of 32, while the three-way classification (HD vs CoV vs Flu) was trained with a smaller total batch size of 8.

## A.2 SUPPLEMENTAL FIGURES

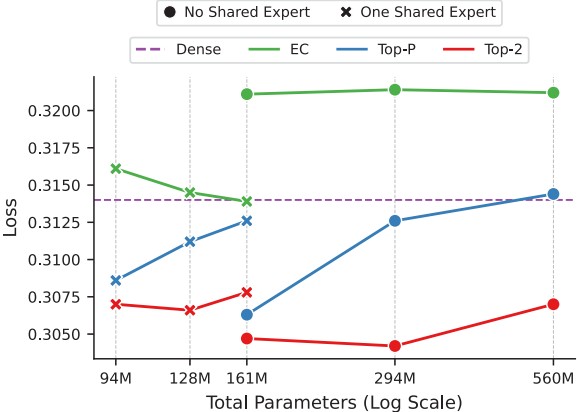

Figure 5: Inference on 100k light chains from the test set, with the models from Fig 1.

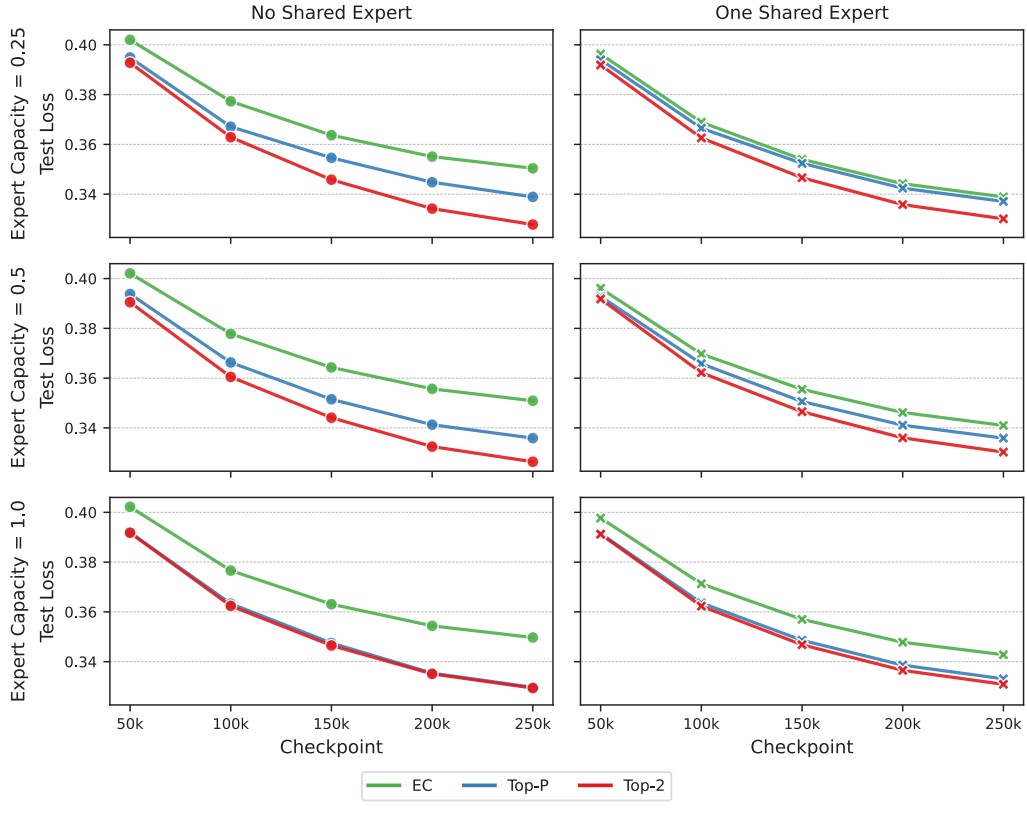

Figure 6: Inference on 100k heavy chains from the test set, at different checkpoints throughout training, with the models from Fig 1.

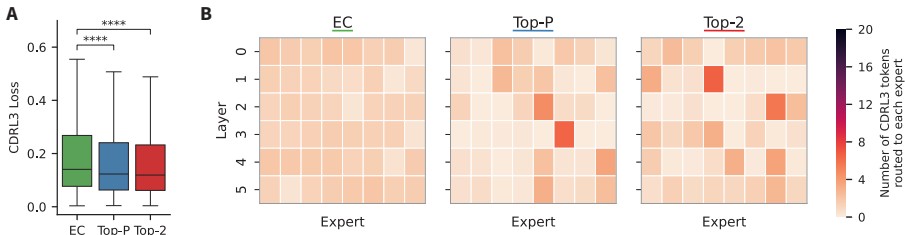

Figure 7: CDRH3 inference and routing of light chains. (A) Per-position inference on 5k light chains from the test set. (B) Routing of the same light chains for each MoE router.

