# OpenReview forum: "Evaluating Expert Specialization in Mixture-of-Experts Antibody Language Models"
_ICLR.cc/2026/Workshop/FM4Science — ICLR 2026 Workshop FM4Science Poster_

### Official Review · Reviewer_jq7z · 2026-02-22
**Evaluating Expert Specialization in Mixture-of-Experts Antibody Language Models**

**Rating:** 7
**Confidence:** 4

**Review:**

### Summary

This paper explores using sparse Mixture-of-Experts (MoE) architectures for Antibody Language Models (AbLMs) to address "germline bias," where models struggle with highly diverse regions like the CDRH3. The authors evaluate different routing strategies, and finds that token-choice routing (Top-K/P) promotes expert specialization in diverse antibody regions, leading to lower loss compared to dense models. They further optimize the router by minimizing the routing of padding tokens, which is essential for training on biological sequences of varying lengths. Their large-scale model, BALM-MoE, outperforms dense counterparts in masked language modeling (MLM) while being significantly more computationally efficient.

### Quality
The technical quality is high, featuring systematic hyperparameter sweeps across expert capacities and shared expert configurations. The experimental framework is rigorous, utilizing paired two-sided t-tests with Bonferroni correction for significance testing. The authors successfully scaled their findings from a 45M active parameter pilot to a 710M total parameter model (BALM-MoE) trained on a massive dataset of 159M unpaired and 2.6M paired sequences.

### Clarity
The manuscript is well-organized. The motivation of using architectural specialization to handle the modular but imbalanced nature of antibody sequences is clearly stated. Figures 1 and 2 effectively illustrate how different routing strategies impact model loss and expert concentration in hypervariable regions. The "Mask Both" strategy for pad-token reduction is well-justified by the data in Figure 3.

### Originality
While MoE is established in NLP, this work provides a novel application tailored to antibody biology. The primary original contribution is the mechanistic insight that token-choice routing enables experts to specifically specialize in the CDRH3 region. Additionally, the domain-specific optimization to ignore padding tokens is a practical innovation that addresses the high-padding reality of mixed paired/unpaired biological datasets.

### Significance
This work is significant for the protein modeling community as it offers a more compute-efficient path for scaling AbLMs. BALM-MoE achieves a 2.5x increase in inference speed compared to a dense model of the same total parameter count. While a slight trade-off in downstream classification accuracy was observed due to smaller embedding dimensions (640 vs 1280), the pre-training gains and efficiency make this a superior architecture for high-throughput screening and potential integration into structure prediction models.

### Pros
Specialization for Diversity: Effectively shows that MoE experts specialize in CDRH3 tokens, improving predictions in the most difficult antibody regions.

Domain-Specific Optimization: The pad-token masking strategy successfully repurposes expert capacity for meaningful biological information.

### Cons
Hyperparameter Limits: The study notes that configurations were modeled on NLP defaults; more antibody-specific tuning (e.g., varying expert sizes) remains for future work.

Structural Gap: While the authors suggest utility for structure prediction, the paper lacks direct structural benchmarks to confirm this.

---

### Official Review · Reviewer_4WA3 · 2026-02-24
**Try to combine MOE with Immunology**

**Rating:** 5
**Confidence:** 3

**Review:**

quality: fair
clarity: very clear. training MOE encoder only model on scientific dataset. comparing with same activated param number dense model
originality: fair
significance: fair

pros: very clear

cons: significance
MOE are usually for training higher-performance model when hitting hardware constraint. "Pilot MoE models were encoder-only models containing 45M active parameters", which doesn't seem to hit hardware constraint.

---

### Meta-Review · Area_Chair_qxc2 · 2026-02-27

**Recommendation:** Accept (Poster)
**Confidence:** 4

**Metareview:**

The average review score is above 6, which means reviewers recommended an acceptance.

---

### Decision · Program_Chairs · 2026-03-03

Accept (Poster)